# Study of the Impact of the Parasitic Microalgae *Coccomyxa parasitica* on the Health of Bivalve *Modiolus kurilensis*

**DOI:** 10.3390/microorganisms12050997

**Published:** 2024-05-15

**Authors:** Ayna V. Tumas, Veronika A. Slatvinskaya, Vadim V. Kumeiko, Yulia N. Sokolnikova

**Affiliations:** 1A.V. Zhirmunsky National Scientific Center of Marine Biology, Far Eastern Branch of Russian Academy of Sciences, 690041 Vladivostok, Russia; 2School of Medicine and Life Sciences, Far Eastern Federal University, 690922 Vladivostok, Russia

**Keywords:** bivalve, microalgae *Coccomyxa*, parasite, hemolymph, immune, histopathology

## Abstract

The invasion of bivalves by parasitic microalgae *Coccomyxa* is widespread and causes pathologies and dysfunctions of the organs, especially in the most valuable products: the mantle and the muscle. The pathogenesis of the disease remains completely unknown. In this study, based on a macroscopic examination of *Modiolus kurilensis* and microalgae count in each infected individual, four stages of disease development with characteristic pathognomonic symptoms were described. During the progression of the disease, the concentration of alkaline phosphatase, glucose, calcium, hemolytic and agglutinating activities, number of basophils, eosinophils, phagocytes, and cells with reactive oxygen species increased in the hemolymph, while number of agranulocytes, cells with lysosomes, dead hemocytes, total protein concentration, as well as the weight of mollusks decreased. In the nephridia and digestive gland, necrosis, invasion of *Nematopsis* sp., hemocyte infiltration, and fibrosis increased. The ratio of changed tubules and occurrence of granulocytomas increased in the digestive gland, while the base membrane, nephrocytes and concretions changed in the nephridia. This study helps establish the variability of these parameters under normal conditions and their alteration during the disease. Moreover, these findings can be used for veterinary monitoring of the state of bivalves in natural and aquaculture populations.

## 1. Introduction

Pathogenic microalgae invading animals are a unique phenomenon in nature. Currently, only two algae species causing diseases have been described and confirmed in terrestrial vertebrates, including humans. These species belong to genera *Prototheca* Krüger, 1894 (Chlorellales: Chlorellaceae) and *Chlorella* Beyerinck, 1890 (Chlorellales: Chlorellaceae), and cause protothecosis and chlorellosis, respectively (these diseases can also be referred as “algaemia”) [1,2,3]. In 2018, the only case of human infection with another microalga, *Desmodesmus* (Chodat) An, Friedl & Hegewald, 1999 (Sphaeropleales: Scenedesmaceae), was reported in Japan [4]. In terrestrial invertebrates, pathogenic *Helicosporidium* spp. Keilin, 1921 (Chlorellales: Chlorellaceae), which only affects insects, has been found and described [5]. Among marine organisms, only bivalves are susceptible to infection by microalgae, from genus *Coccomyxa* Schmidle, 1901 (Trebouxiophyceae ordo incertae sedis: Coccomyxaceae) (*Coccomyxa parasitica* Stevenson & South, 1974 and *Coccomyxa veronica* Sokolnikova, Tumas, Stenkova, Slatsvinskaya, Magarlamov & Smagina, 2022) in particular [6,7]. Unlike any other known cases, there is no name for the disease caused by *Coccomyxa* in mollusks so far due to various reasons. First of all, the biotic status of these microalgae has been debated for a long time, and the negative effects of *Coccomyxa* have only recently been described [6,8,9,10]. The symptoms of infection and the distribution area of *Coccomyxa* are similar to the most destructive pathogens of the genus *Perkinsus* Levine, 1978 (Perkinsida: Perkinsidae) [11,12,13] and affect commercially important species such as *Mytilus edulis* Linnaeus, 1758 (Mytilida: Mytilidae), *Mytilus edulis chilensis* Hupé, 1854, *Mytilus galloprovincialis* Lamarck, 1819, *Mytilus trossulus* Gould, 1850, *Modiolus kurilensis* Bernard, 1983 (Mytilida: Mytilidae)*, Panopea abbreviata* Valenciennes, 1839 (Adapedonta: Hiatellidae) and *Anadara broughtonii* Schrenck, 1867 (Arcida: Arcidae) [6,7,14,15,16,17,18,19]. Studied in this work, *M. kurilensis* is one of the dominant species of zoobenthos in the shallow and infralittoral zones of the Sea of Japan. Therefore, mollusks from genus *Modiolus* are often used as test objects in biomonitoring studies [20,21,22,23,24]. Due to its high biological value, *Modiolus* is also caught as a bycatch for *Crenomytilus grayanus* Dunker, 1853 (Mytilida: Mytilidae) and goes along with it for the production of canned food [25,26].

The pathogenic microalga *Coccomyxa* has a very wide geographic range, infecting mollusks from various regions of the World Ocean with various hydrochemical regimes: from the White and Barents Seas to the coast of Patagonia [6,9,14,15,16,17,19,27,28]. In mollusks that are infected with *Coccomyxa*, the shell becomes significantly deformed, taking on a heart-shaped appearance with gaps forming between the valves and erosions, outgrowths, and pearls forming on the inside of the shell [6,15,16,27,29]. Additionally, the organs become green. In cases of weak infiltration by microalgae, only the hemolymph and posterior part of the mantle and intestine are colored due to them being affected the most [6,16]. However, with massive invasion, all organs are affected, with the most pronounced pathological changes occurring in the mantle, nephridia, and digestive gland [6]. It has been found that there is a widespread infection in local water areas, leading to mollusks experiencing inflammation and histopathology in all organs and tissues [9]. *Coccomyxa* infection also disrupts the process of shell calcification, reduces filtration capacity, and causes changes in metabolic activity and the reproductive cycle. As a result, infected individuals experience significant delays in growth and development, leading to their death in extreme cases [6,7,8,10,15,16,17,19,28].

Currently, data on *Coccomyxa* are sporadic and relate mainly to the symptoms of the disease and description of pathogen cells. However, the development of the disease and the reaction of the host’s immune system in response to the invasion of microalgae have not been studied. It remains unclear why the hemolymph, which plays a vital role in immune reactions, allows the invasion and spread of *Coccomyxa*. Moreover, the immune response that subsequently occurs is not effective.

As reports of commercial bivalve species infection by microalgae continue to increase every year, there is reason to fear the further spread of *Coccomyxa*. This could pose a danger to aquaculture farms in the future [7,9,19]. Studying the disease’s development will provide unique data on the mechanisms of interaction between the immunity of marine animals and the algal pathogens. This information can also be used to carry out veterinary supervision over the condition of aquatic organisms and take preventive measures to eradicate cases of mass infection in mollusks.

The goal of this work was to study the development of disease caused by the parasitic microalga *C. parasitica* in the bivalve mollusk *M. kurilensis*. Based on macroscopic examination and microalgae cell concentration, each infected individual was assigned a disease grade. Also, a thorough analysis of morphophysiological, immunological, and histopathological parameters of mollusks with various degrees of invasion was carried out.

## 2. Materials and Methods

### 2.1. Collection of Animals

Sexually mature specimens of *M. kurilensis* with a shell length of 80–120 mm were collected from Podsobnaya Bay of the Vostok Bay (Peter the Great Bay, 42°53′17.0″ N, 132°43′17.8″ E) of the Sea of Japan from a depth of 1.5–2 m. Among them were healthy and microalgae-infected mollusks with varying degrees of invasion.

### 2.2. Examination of M. kurilensis

Linear shell size (length and width) of the mollusks was measured by a caliper to the nearest 0.05 mm. The age of each individual was determined by counting the growth rings on the external surface of the shell.

### 2.3. Hemolymph Sampling

Hemolymph was collected from the posterior adductor muscle sinus using a 5 mL sterile syringe into a precooled microtube to avoid hemocyte aggregation.

For total hemocyte count, an aliquot of hemolymph (1 mL) was immediately mixed with an equal volume of 4% paraformaldehyde (PFA) solution prepared in artificial seawater (ASW), containing 460 mM NaCl, 9.4 mM KCl, 48.3 mM MgCl_2_·6H_2_O, 6 mM NaHCO_3_, 10.8 mM CaCl_2_·2H_2_O, and 10 mM N-(2-hydroxyethyl) piperazine-N′-2-ethanesulfonic acid (HEPES) (pH 7.5) at an osmolality of 1090 mOsm. The rest of the hemolymph was centrifuged at 800× *g* for 12 min at 15 °C. After centrifugation, the supernatant was frozen at −85 °C for further humoral immunity activity and biochemical analyses, and the hemocyte pellet was resuspended in ASW and used to assess the parameters of hemolymph cellular factors.

### 2.4. Analysis of Cellular Parameters of Hemolymph

#### 2.4.1. Haemocyte Count and Their Viability

After centrifugation, an aliquot of the hemolymph (100 µL) was incubated for 10 min with 0.1% trypan blue at room temperature (RT). The total number of hemocytes and dead cell ratio as well as the concentration of microalgae were counted by a hemocytometer under a Primo Star microscope (Zeiss, Oberkochen, Germany).

#### 2.4.2. May–Gruenwald Typing of Cell Populations

To identify cell types, 15 µL of the cell suspensions were placed on slides and incubated for 20 min at RT for their adhesion, then fixed in 4% PFA, and stained according to May–Grunwald (Sigma-Aldrich, St. Louis, MO, USA). Then, the slides were mounted and observed under a light microscope. Number of hemoblasts, agranulocytes, basophilic, and eosinophilic granulocytes per 200 hemocytes was counted, as previously described by Sokolnikova and colleagues [30].

#### 2.4.3. Phagocytosis Assay

To initiate in vitro phagocytic reaction, a bacterial suspension (*Staphylococcus* sp., strain 636, previously isolated from marine aquatic organisms and stored at −85 °C in the bacterial cultures collection at the A.V. Zhirmunsky National Scientific Center of Marine Biology) stained with a 0.01% solution of fluorescein-5-isothiocyanate (FITC, MP Biomedicals, Santa Ana, CA, USA) was added to the slides with adherent hemocytes at a rate of 15–20 bacterial cells per hemocyte. Reaction was stopped 1 h 20 min after the addition of bacteria by fixation in 4% PFA for 1 h at RT. Than slides were observed under a Axio Imager A1 fluorescence microscope (Zeiss, Germany). Phagocytic activity was calculated as a percentage of hemocytes containing bacteria per 200 hemocytes; the phagocytic index was calculated as the average number of bacteria within one hemocyte. The percentage of hemocytes containing microalgae (per 200 hemocytes) and the average number of microalgae within one hemocyte were also calculated.

#### 2.4.4. Estimation of Reactive Oxygen Species of Hemocytes and Content of Lysosomes

To detect reactive oxygen species (ROS) in hemocytes, the unstimulated nitroblue tetrazolium (NBT) reduction test in formazan was performed. For this purpose, adherent hemocytes were incubated with 0.004% NBT solution prepared in ASW for 30 min at 17 °C and fixed for 1 h in 4% PFA. Then slides were mounted and observed under a fluorescence microscope. As a result, the number of NBT-positive cells containing blue-violet diformazan granules per 200 hemocytes was counted.

To detect lysosomes localization in hemocytes, a single drop of acridine orange (AO; Sigma-Aldrich) solution prepared with ASW (final concentration 10 μg/mL) was placed onto a slide and covered with a coverslip. Then, slides were observed under a fluorescence microscope with a fluorescent filter for FITC (BP 450–490/FT 510/LP 515, Zeiss, Oberkochen, Germany). Intact AO-containing lysosomes were orange. The number of AO-positive cells per 200 hemocytes was counted.

### 2.5. Analysis of Humoral Parameters of Hemolymph

#### 2.5.1. Hemagglutination Reaction

For the hemagglutination assay, a suspension of human erythrocytes was fixed as previously described [31]. The assay was carried out in round-bottom 96-well immunological plates according to the following pattern: 100 µL of TBS1 solution (10 mM Tris-HCl, 150 mM NaCl, 15 mM CaCl_2_, pH 7.5) was placed in each well of the plate, and 100 µL of mollusks’ plasma was placed into the bottom row; then, a series of double dilutions (from 1/2 to 1/2048) were performed. Thereafter, 50 µL of erythrocytes (concentration 6 × 10^6^ cells/mL) was added to each well. The last row of the plate without plasma was used as a negative control. After 2 h of incubation at RT, the results of hemagglutination were examined visually and expressed as −log_2_ of titer. The titer of hemagglutination is the highest dilution, producing visible agglutination of erythrocytes.

#### 2.5.2. Degree of Hemolysis

To analyze hemolytic activity, non-fixed human erythrocytes were incubated with mollusks’ plasma to estimate released hemoglobin concentration in the solution. Microtubes with reaction mixtures containing 50 µL of plasma, 50 µL of TBS1, and 400 µL of the erythrocytes (concentration 6 × 10^6^ cells/mL) were incubated for 1 h with periodic shaking every 15 min. Reaction was stopped by adding 1 mL of TBS3 solution (10 mM Tris-HCl, 150 mM NaCl, 30 mM Na_2_EDTA, pH 7.5). Then microtubes were centrifuged at 300× *g* for 15 min at 4 °C. The collected supernatant was transferred into flat-bottomed 96-well plates. The optical density (OD) was detected by a Bio-Rad iMark plate reader (Bio-Rad Laboratories, Hercules, CA, USA) at a 415 nm wavelength.

#### 2.5.3. Biochemical Analysis

The concentrations of total protein, glucose, phosphorus, calcium, magnesium, iron, and alkaline phosphatase in plasma were measured using a Mindray BS-120 Chemistry Analyzer biochemical analyzer (Mindray, Shenzhen, China) according to the standard settings.

### 2.6. Determination of the Degree of Infection

To assess the degree of microalgae invasion, the mollusks were dissected, and a thorough macroscopic examination of their bodies and shells was carried out. External signs (deformations, erosion, gaps, and the presence of shell pearls; the number of infected organs and their appearance: mucification, friability, color, and intensity of microalgae infiltration) were used for preliminary evaluation of the degree of microalgae infection for each individual.

The soft body weight of each specimen was measured, and tissue fragments of the gonad, nephridia, hindgut, digestive gland, mantle, posterior adductor muscle, and gill were sampled for histological analyses. The remaining parts of the soft body were homogenized. The number of microalgae cells in the homogenate was counted by a hemocytometer under a fluorescence microscope in several fields of view to determine the degree of infection. The histological and homogenate numerical data were compared with macroscopic observations.

### 2.7. Histological Examination

The samples of the tissue fragments of each organ were excised, sliced, and immediately fixed with a 4% PFA for 1 h. The samples were dehydrated through a progressive series of ethanol, infiltrated by xylene, and embedded in paraffin. Serial sections (7.0 ± 0.5 μm) were prepared using a Microm HM-360 rotary microtome (Thermo Fisher Scientific, Dreieich, Germany), stained with hematoxylin and eosin solution, and then mounted. The mounted sections were observed under a microscope (Axio Imager A1, Zeiss). The histopathological state of the organs was estimated based on the work previously published by the authors [6].

Further quantitative analysis was performed only on the digestive gland and nephridia, based on their higher susceptibility to microalgae infection and severe tissue damage. A quantitative evaluation of histopathological changes in the organs was performed using the following parameters: karyopyknosis, hypervacuolisation, necrosis, hemocytic infiltration, fibrosis, and invasions of digestive gland and nephridia; thickness of the basement membrane, hyperplasia, and architecture (shape) of nephrocytes; the number, size, shape, and structure (density) of concretions in the nephridium tubules; and the ratio of tubule types, the granulocytomas, and fibromas of the digestive gland, as previously described by Kumeiko and colleagues [32].

### 2.8. Data Analysis

Statistical analysis was performed using Microsoft Excel 2010 and Statistica 6.0 software packages. The Kolmogorov–Smirnov test showed that the distributions of data do not follow the normal distribution law (*p* < 0.05) for all samples, so the analysis was carried out using nonparametric statistics. To test the significance of the hypothesis about the absence or presence of differences in all parameters between the studied samples, the Mann–Whitney test (a non-parametric analog of the t-test) was used. Correlation analysis was performed using the Spearman correlation coefficient R (*p* < 0.05). All data in this work are presented as the mean value ± confidence interval (95%).

## 3. Results

### 3.1. Stages of Infection Caused by C. parasitica

In Podsobnaya Bay, some mollusks were found without signs of *Coccomyxa* infection, while others had various anomalies in their soft body and shell. A microscopic analysis of the tissues showed that 18.7% of caught mollusks were not infected by the microalgae, despite being from the same location and group. Mollusks infected with parasitic microalgae had varying degrees of infection (Figure 1). Macroscopic examination revealed that in mollusks at the first, “mild” stage of infection (14.6% of the entire sample), only the hemolymph is colored green, and the shell surface at the anterior end becomes smoother, with the outgrowths of the periostracum becoming less pronounced (Figure 1A, stage 1) compared with healthy mollusks (Figure 1A, stage 0, Figure 2A). Microscopic analysis (Figure 1B) revealed that during this stage, in 1 milliliter of hemolymph from mollusks, there were 1.3 to 12.5 million algal cells (Figure 1C), and in the soft body, microalgae were either not found or constituted no more than 12.1 million cells per individual (Figure 1D).

In 25% of the studied mollusks, the second, “moderate” stage of infection was observed; green spots were found at the proximal end of the edge of the mantle and hindgut, which are not normally characteristic of these organs. Additionally, the shell was almost half devoid of outgrowths from the periostracum, and the posterior end became blunter than in that of healthy individuals (Figure 1A, stage 2). The hemolymph contained 1.50 to 41.3 million algal cells (Figure 1C), while the number of microalgae in the soft body varied from 4.04 to 132.6 million cells per individual (Figure 1D).

At the third, “severe” stage of infection (25.0% of the entire sample), the proximal end of the edge of the mantle and gonad as well as the siphon and hindgut became green (Figure 1A, stage 3). Pearls were also found in the body. The outgrowths of the periostracum practically disappeared, and the shell became thinner and more rounded. Erosions and outgrowths appeared on its inner side (Figure 2B,D). The number of algal cells varied from 42.50 to 81.87 million cells per milliliter in the hemolymph (Figure 1C) and between 180.00 and 373.10 million cells in the soft body (Figure 1D).

The remaining individuals (16.7%) were assigned to the fourth “critical” stage, where the shell anomalies found in the previous group of mollusks were more pronounced and widespread (Figure 2C). Here, most organs (over 50% of the mantle, posterior adductor muscle, gills, gonad, siphon, and hindgut) were intensely green, friable, and heavily mucused (Figure 1A, stage 4). The number of algal cells varied from 83.13 to 213.13 million in the hemolymph (Figure 1C) and between 393.13 million and 1.33 billion cells in the soft body (Figure 1D).

Correlation analysis showed a high direct correlation (0.99–1.00) between the number of microalgae in the hemolymph and the homogenate (Spearman coefficient, *p* < 0.05). Therefore, for further analysis, we used data on the microalgae content in the hemolymph.

### 3.2. Morphometric Characteristics

As the infection progressed, the mollusks showed a significant (*p* < 0.05) decrease in body weight (R = −0.40) (Figure 2H, Table 1). Additionally, there was a reduction in shell length (R = −0.45) (Figure 2E, Table 1). Although the width of the shell did not change (Figure 2F), this resulted in the shell becoming more rounded (R = 0.53) (Figure 2G, Table 1).

### 3.3. Hemolymph Parameters

In the individuals infected with microalgae, there was a significant (*p* < 0.05) increase in the number of basophils (R = 0.26), eosinophils (R = 0.26), and cells containing ROS (R = 0.54), but a decrease in the number of hyalinocytes (R = −0.57), cells with lysosomes (R = −0.49), dead hemocytes (R = −0.70), and the phagocytic index (R = −0.63) was recorded (Table 1; Figure 3 and Figure 4). The concentration of total protein (R = −0.30) in the mollusks’ plasma decreased, while hemolytic (R = 0.37) and agglutinating activity (R = 0.42) as well as the concentration of alkaline phosphatase (R = 0.30), glucose (R = 0.28), calcium (R = 0.27), and iron (R = 0.26) increased significantly. There were no observed changes in THC, phagocytic activity towards bacteria, and magnesium and phosphorus concentrations.

The total number of circulating hemocytes did not change significantly in infected individuals (1.35 ± 0.10 million cells/mL) compared to healthy ones (1.32 ± 0.13 million cells/mL) (Table 1, Figure 3), but the number of dead cells decreased significantly as the infection progressed (from 18.88 ± 5.26% in healthy individuals to 8.87 ± 1.94% in individuals with stage 4 infection) (Table 1, Figure 3). Additionally, starting from stage 2, the content of agranulocytes in the hemolymph significantly decreased (from 6.95 ± 1.61% to 3.61 ± 1.14%), while the content of basophils (from 33.43 ± 3.24% to 41.55 ± 3.27%) and eosinophils (starting from stage 3) (from 48.56 ± 4.82% to 53.31 ± 3.62%) significantly increased in the hemolymph (Table 1, Figure 3).

The number of cells containing ROS and lysosomes remained unchanged at the beginning of infection (proportion of cells with ROS: 34.66 ± 4.87%, with lysosomes: 36.50 ± 1.93%), compared to healthy individuals (proportion of cells with ROS: 29.11 ± 7.85%, with AO: 36.31 ± 1.50%). But this number significantly increased starting from stage 2 (52.23 ± 9.40%), reaching 52.53 ± 6.99% at stage 4, whereas the proportion of cells with lysosomes decreased significantly (from 30.57 ± 2.18% to 28.06 ± 2.06%) (Table 1, Figure 3).

The concentration of total plasma protein significantly decreased by stage 4 of infection (from 0.76 ± 0.11 to 0.56 ± 0.08 mg/mL). Hemolytic activity significantly increased from the very beginning of infection (healthy—47.48 ± 16.85%, stage 1—82.42 ± 8.47%, stage 4—77.45 ± 6.70%), and hemagglutinating activity (healthy—10.33 ± 4.05 −log_2_, stage 2—32.58 ± 3.85 −log_2_, stage 4—28.38 ± 5.58 −log_2_) and glucose concentration (healthy—6.06 ± 0.01 mmol/L, stages 2 and 4—0.09 ± 0.01 mmol/L) significantly increased from stage 2. The content of calcium (17.2–24 mmol/L), alkaline phosphatase (0.81–21 IU/L), and iron (0–29 mmol/L) had similar dynamics of increase, but with less pronounced changes, due to wide variations of these parameters within each animal group (Table 1, Figure 4).

### 3.4. Histopathological State of the Digestive Gland and Nephridia

Upon analyzing the histopathological condition of the organs and tissue of *M. kurilensis*, it was found that the mantle, digestive gland (DG), and nephridia showed the most pronounced histopathological changes (Figure 5). In all reaction patterns, histopathological alterations in the DG and nephridia of the infected bivalves were significantly higher compared to the healthy ones (Table 2).

The development of microalgae infection in mollusks caused changes in the DG, mainly affecting the interstitial space (IS). There was a significant (*p* < 0.05) increase in the frequency and necrotic area (R = 0.75) (Figure 6C), infiltration of IS by hemocytes (R = 0.68), IS area containing fibrillar tissue (R = 0.69), and the number of parasites *Nematopsis* sp. Schneider, 1892 (Eugregarinorida: Porosporidae) in IS (R = 0.37) (Figure 6B,D). Cells with pyknotic nuclei appeared in the tubes (R = 0.29), the number of type I digestive tubes themselves (holding) decreased (R = −0.22), and the number of type IV (recovering) tubes increased (R = 0.35) (Figure 6F). Changes in these parameters were mainly logarithmic in nature (with a predominance of significant differences at stages 2–3 of infection development) except for fibrosis, which was characterized by linear growth (from 12.81 ± 3.72% in healthy individuals to 77.19 ± 11.36% in stage 4). In healthy mollusks, DG did not have cells with pyknotic nuclei, areas of necrosis and granulocytomas (Figure 6A), and parasites and hemocytes were practically absent in the IS (Table 2, Figure 7).

In the nephridia, significant (*p* < 0.05) histopathological changes were present in both the tubules and the IS. In the nephridia IS, with an increase in the number of microalgae in the hemolymph, the following significantly increased: infiltration with fibrillar tissue (R = 0.68) (Figure 8B), the number of hemocytes (R = 0.41) (Figure 8B), and the frequency of occurrence of parasites *Nematopsis* sp. (R = 0.33). In the tubes, the membrane thickness (R = 0.40) (Figure 8D), the frequency of occurrence and area of necrosis (R = 0.46) (Figure 8D), and cases of changes in epithelium shape from cylindrical to round (R = 0.36)—due to an increase in its width (R = 0.15) (Table 2, Figure 9)—increased as well. The most significant changes were observed in concretions (Table 2, Figure 8C,D), concerning all their parameters: shape (R = −0.26), density (R = −0.32), size (R = 0.17), and number (R = 0.25). Similarly to DG, the changes in nephridia parameters were logarithmic, except for the size of large concretions (exponential growth) and epithelial circumference (linear growth) (Figure 9).

The presence of algae in the mollusks did not affect the incidence of hyperplasia and the height of nephridium nephrocytes (Figure 9), and in the DG, it did not affect the incidence of hypervacuolated cells and granulocytes (Figure 6B).

## 4. Discussion

Currently, aquaculture is facing a major economic challenge due to the periodic occurrence of epizootics [33,34,35,36,37]. This is caused by the lack of knowledge describing mechanisms of pathogen–host interactions, which limits the methods of diagnosis and treatment. Pathogens of bivalves include a wide range of organisms, with protozoa posing the greatest threat [38,39]. Green microalgae belonging to the genus *Coccomyxa* (*C. parasitica* and *C. veronica*) are the first microalgae whose dangers are not toxins. *Coccomyxa* invades mollusks, causing serious pathologies. Although microalgae of the *Coccomyxa* genus have been found in *P. abbreviata* [17] and *A. broughtonii* [7]; this parasite mainly affects members of the Mytilidae family.

It has been found that *Coccomyxa* infections in Mytilidae family members cause microalgae infiltration in all organs, but with a predominance in the mantle, gonad, nephridia, and DG [6,7,8,9,15,17,27,28,29]. This invasion leads to severe pathological changes, such as abnormal histomorphological changes and organ dysfunction, accompanied by pronounced inflammatory response and tissue degeneration. The affected organs show deformation of the base membrane, changes in the epithelium, hemocyte infiltration, and fibrillar tissue in the interstitial space. This study found that the mantle, DG, and nephridia of the *M. kurilensis* were the most affected organs. Additionally, hemocyte infiltration and epithelial changes were also observed in the intestine, posterior adductor muscle, and gills, which are the first to be invaded by microalgae and were studied in detail for the first time in this research. This study also found that mucoid epithelial cell activity increased in the mantle, but the number of mucoid cells decreased in the gills, possibly due to the abundant secretion of mucus as a specific protective reaction of the body [40]. In the intestine, there were observations of vacuolation of the epithelium, membrane dilatation, and dilation of algae-containing sinuses. In addition, in the posterior adductor muscle, pearls and the loosening of fibers detectable without any equipment were observed along with atrophy and dilatation of the perimysium that were detected microscopically. In all the above-mentioned organs, encapsulation of algae was commonly observed with the formation of large fibromas/granulocytomas and areas with predominantly necrotic material. Similar findings were reported in other works about microalgal invasions [15,16,17,27,29] and in studies examining the influence of parasites on the physiological state of mollusks [41,42,43]. For example, hemocytes’ extensive encapsulation of *Perkinsus chesapeaki* McLaughlin, Tall, Shaheen, El Sayed & Faisal, 2000 cells were observed in *Mya arenaria* Linnaeus, 1758 (Myida: Myidae) and *Tagelus plebeius* Lightfoot, 1786 (Cardiida: Solecurtidae) [44]. A study conducted in 2012 observed a similar pattern of response in *Ruditapes philippinarum* Adams & Reeve, 1850 (Venerida: Veneridae) and *Ruditapes decussatus* Linnaeus, 1758 [45]. In *Tridacna crocea* Lamarck, 1819 (Cardiida: Cardiidae), infected by *Perkinsus olseni* Lester & Davis, 1981, various symptoms such as inflammation and necrosis of the visceral mass and gills, atrophy of glandular, gonadal, and muscular tissues, and partial distension of the mantle were detected [46]. *Coccomyxa* also shows differences in disease expression. Thus, unlike mussels, in the geoduck *P. abbreviata*, which burrows into the ground, the infiltration of hemocytes and microalgae cells was detected only in the hindgut and siphon, which distorted the orientation of muscle fibers due to the dense accumulation of pathogens [17]. In contrast, in the *Anadara* Gray, 1847, with a similar lifestyle to the geoduck, pathologies comparable to mythilids were observed [7].

There are several serious diseases that affect bivalves, especially oysters. One of these diseases is bonamiosis, which is caused by protozoa of the genus *Bonamia* Pichot, Comps, Tigé, Grizel & Rabouin, 1980 (Haplosporida: Haplosporidiidae). These protozoa colonize only the host hemocytes, except for *Bonamia perspora* Carnegie, Burreson, Hine, Stokes, Audemard, Bishop & Peterson, 2006, which lives in connective tissue [36]. They mainly invade the gills [37] and nearby connective tissues, thereby avoiding digestion by the host [47]. However, recent research by Lane and colleagues (2016) has shown that the majority of infected hemocytes are also present in the connective tissue of the interstitial space between the tubules of the DG and the acini of the gonads [48]. They also form aggregates in the marginal zone of the mantle, which can cause yellowing of the mantle, as well as extensive damage to the gills and connective tissue [36]. Another disease that affects oysters is martheliosis, which is caused by two protozoans from the genus *Marteilia* Grizel, Comps, Bonami, Cousserans, Duthoit & Le Pennec, 1974 (Paramyxida: Marteiliidae): *Marteilia refringens* Grizel, Comps, Bonami, Cousserans, Duthoit & Le Pennec, 1974 and *Marteilia sydneyi* Perkins & Wolf, 1976, that were identified from *Ostrea edulis* Linnaeus, 1758 (Ostreida: Ostreidae) in France and *Saccostrea glomerata* Gould, 1850 (Ostreida: Ostreidae) in Australia in the late 1960s [49]. Both pathogens affect the DG, forming spores in epithelial cells, causing whiteness and necrosis of the gland. As a result, mollusks become lean, stop growing, and eventually die [36]. *Haplosporidium* Lühe, 1900 (Haplosporida: Haplosporidiidae) is another pathogen that affects oysters and mussels, causing MSX (Multinucleated Sphere Unknown) disease [37]. In infected mussels, the parasite is usually found in the connective tissue and sinuses of the visceral masses, with spores found in the epithelium of the DG tubules. In oysters, spores were also present in the mantle, gonad, and gills besides the DG. However, the major site of hemocyte infiltration was found to be the gills [50].

Thus, the above-mentioned studies show that the primary area of infection by pathogens in mollusks are the DG and hemolymph. These are the crucial systems that are responsible for metabolism and the elimination of unwanted agents. Therefore, our focus in this research was to diagnose and evaluate the condition of these systems during the development of infection caused by *Coccomyxa*. We also studied the nephridia, which are vital organs involved in the processes of transformation and detoxification.

After conducting a thorough macro- and microscopic analysis of the tissues and organs of the bivalve mollusks *M. kurilensis* infected with the green parasitic microalgae *C. parasitica*, we have identified four stages of disease development. As the disease progressed, the infiltration of the tissues and organs of the mollusks by microalgae increased. However, at the initial stage, there were no visible changes in the mollusks, and microalgae were only detectable through microscopic examination. As microalgae infiltration increases, changes in the mollusks’ shells became evident, especially from second stage. The outgrowths of the periostracum disappeared (“baldness”), and the proximal end of the valves thickened and rounded, producing a more rounded or heart-shaped shell. At stage three, due to calcite deposits, erosion and outgrowths, the shell was severely deformed. Studies examining microalgae infestation of mussels have also observed severe shell erosion and uneven growth, especially at the posterior end [10,15,16,27,51]. The authors point out that most of the shells also lost a significant part of calcite, exposing part of the pearl [27,28]. This allowed the algae living in the tissues to receive enough light for photosynthesis. Some authors revealed large quantities of pearls in the mantle tissues of infected individuals upon autopsy, and tissues located in the posterior part of the body acquired a green coloration [27]. As the invasion progressed to stage three or four in *M. kurilensis*, shell anomalies became more severe, and the organs of the proximal part of the body acquired a more pronounced green color. Caceres-Martinez and colleagues have described similar developments in disease caused by the protozoan parasite, *Perkinsus* [52]. In mild infections, *Perkinsus* cells were observed in the intestinal epithelium, whereas in severe infections, infiltration of hemocytes and *Perkinsus* was observed in connective tissue, DG, mantle, gills, and gonads [52]. In another study, *Perkinsus* sp. were found in the gill plates of weakly infected *M. arenaria*, whereas in heavily infected individuals, an increase in the number and size of hemocyte-encapsulated granulocytoma cysts was also observed in the gills, interstitial space between the tubules of the DG, gonads, and nephridia, which often contained free and encapsulated *Perkinsus* cells [53]. Additionally, the development of *Perkinsus* infection was usually accompanied by blanching of the mantle [54,55], while with *Haplosporidium* invasion, the mantle became thin, watery, and brownish color [50]. A change in mantle color (to yellow) has also been observed in individuals affected by *Bonamia* [36]. However, these studies did not quantitatively analyze the number of parasites and their impact on the condition of mollusks, which makes it impossible to assess the degree of irreparability of the changes caused by the parasite, the host’s immune system, or the possibility of treating mollusks. This is particularly important for aquaculture farms. It has been established that if a mollusk does not die from *Perkinsus* infection, then its fatness index is significantly reduced, and its storage tissues and gametes are also reduced [56]. In this study, we also observed a significant decrease in the soft body weight of bivalves, which is the most valuable product in this species of mollusks.

To evaluate the condition of the DG of the *M. kurilensis*, besides a qualitative analysis to determine the severity of pathological abnormalities, a quantitative analysis was performed on eight parameters that are commonly used in biomonitoring and assessing pathological abnormalities [32,41,57,58,59,60,61,62,63,64,65]. It was observed that the *Coccomyxa* invasion in *M. kurilensis* led to an increase in the infiltration of hemocytes and fibrillar tissue into the interstitial space of the DG, which resulted in the formation of granulocytomas and fibromas. This indicates a significant inflammatory response of the host’s body to the presence of microalgae. Studies have shown that even minor disturbances in the body of mollusks can cause a change in the characteristics of hemocytes, their mobilization, and migration to the organ experiencing the greatest load from the damaging factor [66,67]. For instance, in *Macoma balthica* Linnaeus, 1758 (Cardiida: Tellinidae), severe hemocytic infiltration of the DG was observed during the initial stage of *Perkinsus* infection [44,68]. Similarly, in oysters, hemocyte infiltration, and the presence of *Perkinsus* cells in the connective tissue, digestive epithelium, and mantle were also noted [52,68]. After exposure to the harmful algae *Prorocentrum minimum* (Pavillard) Schiller, 1933 (Prorocentrales: Prorocentraceae), hemocyte diapedesis, pathogen isolation in the intestinal lumen were recorded in *M. edulis*, which led to an increase in the number of bacteria in the stomach and intestines. The migration of hemocytes to these organs contributed to the severe inflammatory processes [69].

It is worth noting that in mollusks with a high degree of infection with microalgae, there is a progressive change in the level of necrosis. Additionally, there is a decrease in the number of digestive tubes of type I (holding) and an increase in type IV (recovering). This indicates that destructive processes are occuring in the DG and its functions are being disrupted. In 2017, mollusks infected with *Perkinsus* were examined, and it was found that the pathogen, at terminal stages, causes hemocytosis and lysis of affected tissues [36]. Furthermore, it leads to necrosis of the epithelial cells of the DG [52,68]. Therefore, our findings suggest that changes in DG are critical and irreversible. Due to metabolic disorders, these changes can not only cause a delay in the development and growth of infected mollusks but can also lead to their death.

The nephridia plays an important role in maintaining the body’s physicochemical constants and regulating the metabolism of organic and inorganic substances. They also store and excrete metabolic products. However, the nephridia are more selective to certain types of irritants, which makes them less useful as a bioindicator of alterations in the condition of mollusks and their habitat [70]. Changes in concretions in the nephridia are mainly evaluated to detect various pollutants [71,72]. The nephridia are also common sites for *Haplosporidium* invasion and the larval stage of trematodes, which cause strong inflammatory reactions resulting in granulocyte formation [73]. *Coccidia* Leuckart, 1879 (Alveolata: Apicomplexa) parasites can be found inside the nephrocyte or in the lumen of the renal tubule, causing hypertrophy of epithelial cells [41]. Hypertrophy, as well as severe infection, can lead to the death of artificially grown mussels and scallops [74]. In 2023, a new apicomplexan parasite infection (provisionally named BSM) was discovered in *Argopecten irradians* Lamarck, 1819 (Pectinida: Pectinidae); it caused deformation or loss of integrity of the renal tubules and detachment or destruction of nephrocytes, but did not cause noticeable hemocyte infiltration [75]. In mollusks infected with *Coccomyxa*, we have observed the destruction of nephrocytes in the nephridia. This led to the accumulation of a large number of concretions with an irregular shape, causing a change in their morphology, making them more rounded. While the size of the concretions changed significantly with the development of infection, their density significantly decreased as well. Furthermore, with the appearance of microalgae in mollusks, the base membrane in the tubules thickened sharply (even at the 1st stage of infection). This thickening could serve as a barrier to further penetration of the parasite. Similarly to DG, changes were observed in the nephridia of infected individuals with increased infiltration by microalgae, namely, an increase in the number of hemocytes and fibrillar tissue area in the interstitial space, as well as necrosis. A similar pattern of necrosis was observed during viral infection of *Atrina lischkeana* Clessin, 1891 (Ostreida: Pinnidae) [76]. No other histomorphological changes in bivalves relative to the nephridia have been studied.

Our research discovered that mollusks infected with *Coccomyxa*, showed parasitic infestation of *Nematopsis* sp. in both DG and nephridia, which increased significantly with the severity of *Coccomyxa* infection. The virulence of a pathogen depends on the state of the host’s defense system, and the appearance of a concomitant infection may indicate the suppression of the host’s defense system by microalgae, resulting in a weakened immune system.

In order to understand the factors that lead to the development of *Coccomyxa* invasion in mollusks, we analyzed the state of the cellular and humoral factors of the hemolymph of the *M. kurilensis.* This is considered a key line of defense for bivalves. Our findings show that during the initial stages of invasion, the organism attempts to eliminate the antigen by actively phagocytizing microalgae cells. However, the more active the phagocytic activity in mollusks, the more actively the infection develops. This indicates that the protection system of mollusks against these microalgae is probably not effective and only promotes dissemination. We observed a decrease in the number of lysosomal cells but an increase in granulocytes. This may be due to either the oxyphilicity of *Coccomyxa* or its ability to induce a decrease in lysosomal membrane stability [77]. It was also discovered that *Bonamia ostreae* Pichot, Comps, Tigé, Grizel & Rabouin, 1980 can disrupt the process of phagocytosis, causing the destruction of lysosomes, and suppress the synthesis of ROS [78]. Vibrios *Vibrio splendidus* (Beijerinck, 1900) Baumann, Baumann, Bang & Woolkalis, 1981 (Vibrionales: Vibrionaceae)*, Vibrio tasmaniensis* Thompson, Thompson & Swings, 2003, and *Vibrio aestuarianus* Tison & Seidler, 1983 in mollusk hemocytes caused a sharp decrease in the stability of lysosomal membranes or disrupted the PI3-K signaling pathway that regulates phagosome-lysosome fusion. 

After the microalgae are absorbed and ineffectively killed by a set of lysosome enzymes of circulating phagocytes, there is a slight increase in the number of hemocytes and large-scale oxidative stress is triggered in the hemolymph. This is evidenced by an excessive increase in NBT test values during the second stage of infection, which leads to only a minor phagocytosis of microalgae. According to the literature, representatives of the genus *Coccomyxa* have an effective antioxidant system as lutein [79,80], carotenoids (such as alloxanthin, diadinoxanthin and diatoxanthin, including in *C. parasitica* [27,29]), fatty acids (such as linolenic acid in *Coccomyxa* sp. (strain *onubensis*) and *Coccomyxa gloeobotrydiformis* Reisigl 1969 [81]), and enzymes such as glutathione reductase, ascorbate peroxidase, and catalase [82]. These compounds protect microalgae from the harmful effects of ROS. Soudant and colleagues (2013) showed that *P. marinus* can inhibit ROS production or remove ROS products due to the presence of acid phosphatase (O_2_^−^ inhibition), superoxide dismutase (O_2_^−^ neutralization), and ascorbate peroxidase (peroxide scavenging) [83]. Inhibition of at least one of these leads to a noticeable increase in ROS production and apoptosis of infected hemocytes, resulting in the death of the pathogen [84]. However, the role of specific forms of ROS in cell self-destruction and the mechanisms of cytotoxicity are not completely clear. Moreover, there is no simple answer to the question of whether oxidative stress is a consequence or an inducer of functional changes that accompany the development of programmed cell death, despite the obvious relationship between oxidative stress and apoptosis. It is possible for microalgae to directly induce apoptosis through the toxins they produce. *Coccomyxa*, for example, may inhibit cell death, allowing it to evade the host’s defense system and spread throughout the host’s body. This could explain the increase in phagocytes containing microalgae when hemocytes’ phagocytic activity is low. Both cellular and humoral factors are involved in the active response to the infection. For example, there is a significant increase in alkaline phosphatase, hemolytic, and agglutinating activity of plasma during the 1st stage of infection, but a gradual decrease in the total protein concentration (likely due to protein adhesion to algae). In an experiment by with *Chlamys farreri* Jones & Preston, 1904 (Pectinida: Pectinidae) were infected with *Listonella anguillarum* (Bergeman 1909) MacDonell & Colwell 1986 (Vibrionales: Vibrionaceae), humoral factors increased, but no antibacterial or agglutinating activity was observed [85]. In the eastern oyster *Crassostrea virginica* Gmelin, 1791 (Ostreida: Ostreidae) infected with *P. marinus*, there was a significant decrease in plasma protein concentration, while in *Crassostrea gigas* Thunberg, 1793 plasma agglutinating activity increased (with higher agglutinin titer in severe infection in *C. virginica* than in *Crassostrea gigas*) [86]. In *R. decussatus* severely infected with *Perkinsus atlanticus* Azevedo, 1989, the titer of agglutinins was significantly higher, but the antimicrobial activity of plasma was lower in comparison to healthy mollusks [87]. One study showed that hemolymph agglutinin activity increased 2 days after incubation of *C. gigas* in a medium with *Vibrio anguillarum* Bergeman, 1909, which persisted up to 7 days [88].

During infection, infected hemocytes that fail to clear *Coccomyxa* may self-destruct (e.g., by apoptosis) to reduce the spread of the invading organism. However, in the mollusks at the third and fourth stages of infection, the number of dead cells decreases. This may be because microalgae either inhibits the apoptotic pathway through signaling molecules (due to high levels of ROS) so that the host does not eventually die or the host cells adapt to the invader. Programmed cell death plays a significant role in disease progression and the destruction of “harmful” cells, and its inhibition is one strategy for strategies for successful survival of the parasite [89]. It has been established that the highly virulent strain of *P. marinus* modulates apoptosis of oyster hemocytes, promoting the development of infection. At the initial stages of infection, an increase in apoptosis occurred in vitro and in vivo, followed by suppression and a return of apoptosis to the initial level 8–24 h after infection, indicating parasite-induced inhibition of the immune response. During infections with intermediate- or low-virulence strains of *P. marinus*, a temporary suppression of apoptosis was observed 4–8 h after infection, and at later stages, a sustained increase in hemocyte apoptosis was found. This indicates that the hemocytes were able to overcome parasitic suppression of this programmed cell death and successfully fight the infection [90]. Studies on the mechanisms of apoptosis caused by *P. marinus* have shown that early post-infection stimulation of apoptosis is not caspase-dependent but occurs through the mitochondrial pathway (Bcl-2, anamorsine) [84]. However, in 2022, Witkop and colleagues reported that in the eastern oyster, infection with *P. marinus* suppresses in vitro apoptosis through IAP-dependent (inhibitor of apoptosis) pathways [91]. When shellfish were infected with *B. ostreae*, the remodulation of apoptosis occurred with the participation of genes such as IAP and apoptosis factor (AIF) [92]. The data obtained on the immune response of *M. kurilensis* to *Coccomyxa* will allow for more definite studies in the future to understand the mechanisms of evasion and resistance of these pathogens, as well as the weakness of the host’s immune system. This weakness allows the pathogen to infect it and thrive. Knowledge of these mechanisms will make it possible in the future to carry out immune-corrective manipulations in aquatic organisms to enhance their protection against pathogenic microalgae.

Biochemical analysis of mollusk plasma showed that the concentrations of glucose, calcium and iron in the plasma changed significantly in infected mollusks compared to healthy ones. As the *Coccomyxa* infection progresses, the level of iron in the hemolymph of mollusks increases. This may be a response to an increase in ROS, as iron often acts as a cofactor for antioxidant enzymes. The increase in glucose may be a consequence of the transport of this metabolite from microalgae. Similar changes were observed in the mollusks *Tridacna gigas* Linnaeus, 1758 (Cardiida: Cardiidae) and *T. crocea*, which form a symbiosis with zooxanthellae *Symbiodinium* sp. Freudenthal, 1962 (Suessiales: Symbiodiniaceae) [93,94]. It has been shown that representatives of the genus *Coccomyxa*, being symbionts of a number of terrestrial lichens [95], supply them with the ribitol, which supports the vital activity of fungi [96]. The calcium content in the hemolymph could increase for several reasons. Firstly, Srednyaya Bay, from where the mollusks were caught, is a eutrophicated water area, in which the phosphorus content exceeded normal levels [97,98], and in some years, an increase in the number of species and density of green microalgae was observed [99]. Bivalves have been shown to actively absorb phosphorus from the environment [100,101], remineralizing it and converting it into soluble reactive phosphorus (SRP, a bioavailable form) [102]. A shift in the ratio of the concentration of phosphorus and calcium in the hemolymph can, in turn, cause the leaching of the ions from the structures of the body containing it (for example, shell). Thus, it was shown that shell dissolution caused an imbalance of calcium in hemocytes and reduced the need for the absorption and transport of calcium in the tissues of the gills and mantle in oysters [103]. In *M. galloprovincialis, M. edulis* and *C. gigas*, an increase in hemolymph Ca^2+^ content due to shell dissolution of CaCO_2_ to compensate for acidosis has been reported [104,105,106]. The work of Thomsen and Ramesh revealed that calcification in mussel larvae is strongly linked to calcium concentration levels in water and the extrapallial fluid [107]. In addition, in 2018, Zuykov and colleagues suggested that endolithic cyanobacteria, found in abundance in *Coccomyxa*-infected mollusks, also make a significant contribution to shell erosion, which may also be the cause of increased calcium levels in plasma of the *M. kurilensis*. *Coccomyxa* has wide adaptive potential, inhabiting various biotopes and organisms, so under conditions of a changing climate with acidification and eutrophication of water areas, there is a possibility of a wider distribution of this pathogen [9]. Moreover, in the past ten years, ten new species of *Haplosporidia* have been discovered since the last review by Burreson and Ford [108]. Our research on the immune response in *M. kurilensis* reveals the high resistance of the pathogen to the host’s protective reactions. This resistance may be due to the evolutionarily undeveloped mechanisms of the immune response in mollusks to photoautotrophic microorganisms. Further research on the morphological and molecular structure of microalgae may help us understand the reasons for the high pathogenicity of *Coccomyxa* and aid in the fight against this pathogen. To comprehend the cause of the susceptibility of aquatic organisms to *Coccomyxa*, simulated experiments must be conducted on the infection of shellfish in laboratory conditions. These experiments should modulate various factors relating to the state of the shellfish (suppression of the immune system, injury, weakening, etc.) and the environment (acidification, excess minerals, etc.), which will further carry out preventive anti-epizootic measures.

## 5. Conclusions

The high mortality rate of bivalves due to various invasions is a major issue in aquaculture that leads to significant economic loss in the industry. Unfortunately, knowledge about the defense responses of bivalves is still limited, and pathogen control relies mainly on preventive strategies and removal of diseased individuals. One of the latest pathogenic microorganisms identified in a wide range of commercial mollusks is a green microalga of the genus *Coccomyxa*. This pathogen causes serious organ pathologies in infected individuals, leading to disruption of their functions. In mollusks infected with *C. parasitica*, an increase in the frequency and intensity of infiltration of the interstitial space with fibrillar tissue and hemocytes as well as necrosis and parasites was observed in the digestive gland and nephridia. Furthermore, there was a change in the parameters of the concretions and base membrane in the nephridia. In the DG, the ratio of the digestive tubes changed. Although infected mollusks showed a significant increase in the hemolytic and agglutinating activity of plasma, the number of hemocytes and the proportion of granulocytes, as well as their phagocytic activity and ROS synthesis compared to healthy individuals, the entire complex of response reactions aimed at destroying the pathogen turned out to be ineffective. This could probably be due to the presence of special mechanisms in *Coccomyxa* that evade the host’s immune defense. The identified pathogenetic changes in infected mollusks indicate an extremely negative effect on the physiological state of the *M. kurilensis*. This not only leads to a decrease in their health level but also increases susceptibility to concomitant diseases, including those caused by parasites other than *Coccomyxa*. The obtained data on the limits of variability of hemolymph parameters in normal conditions and their changes against the background of disease progression can be used in the future for veterinary control and monitoring the condition of aquatic organisms in natural and aquaculture populations. The constructed dynamic model of the development of invasion in *M. kurilensis* with a description of the symptoms specific to this infection will enable us to take necessary measures in advance to limit the spread of the pathogen.

## Figures and Tables

**Figure 1 microorganisms-12-00997-f001:**
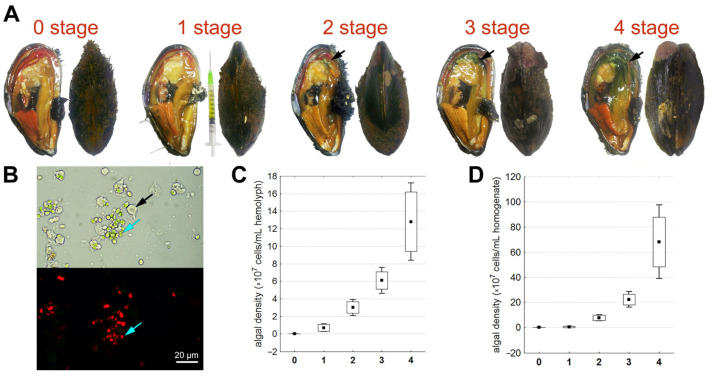
Assay of the *C. parasitica* infection in the *M. kurilensis*. (**A**) Soft body and shell alterations during the progression of infection (black arrowheads indicate microalgae-infiltrated areas). (**B**) Microalgae (blue arrow) among hemocytes (black arrow) in a native hemolymph under light and a fluorescent microscope. (**C**) *Coccomyxa* cell concentration in the hemolymph in mollusks with varying degrees of infection. (**D**) *Coccomyxa* cell concentration in the soft body homogenate in mollusks with varying degrees of infection. The values are presented as mean ± standard deviation.

**Figure 2 microorganisms-12-00997-f002:**
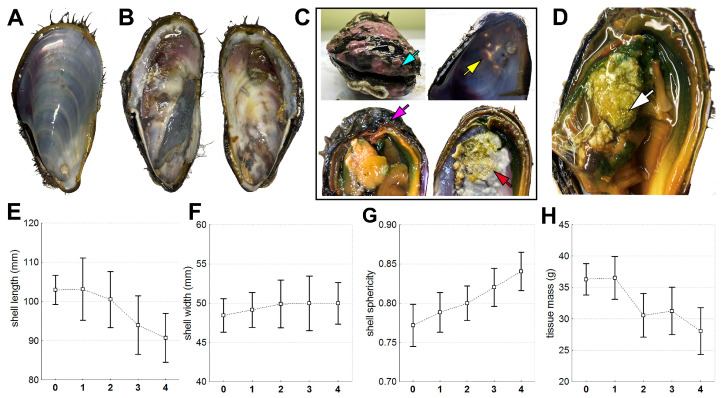
Body and shell changes during the progression of infection. (**A**) Inner surface of the shell of a healthy individual. (**B**) Inner surface of the shell of an infected individual (stage 3). (**C**) Shell alterations at stage 3 and 4 of infection (blue arrow—disappearance of periostracum outgrowths, gap between the valves; yellow arrow—shell thinning; purple arrow—wrapping of the shell edge; red arrow—erosion). (**D**) Shell pearls in the muscle (white arrow). (**E**–**G**) Shell parameters. (**H**) Soft body mass. The values are presented as mean ± 95% confidence interval.

**Figure 3 microorganisms-12-00997-f003:**
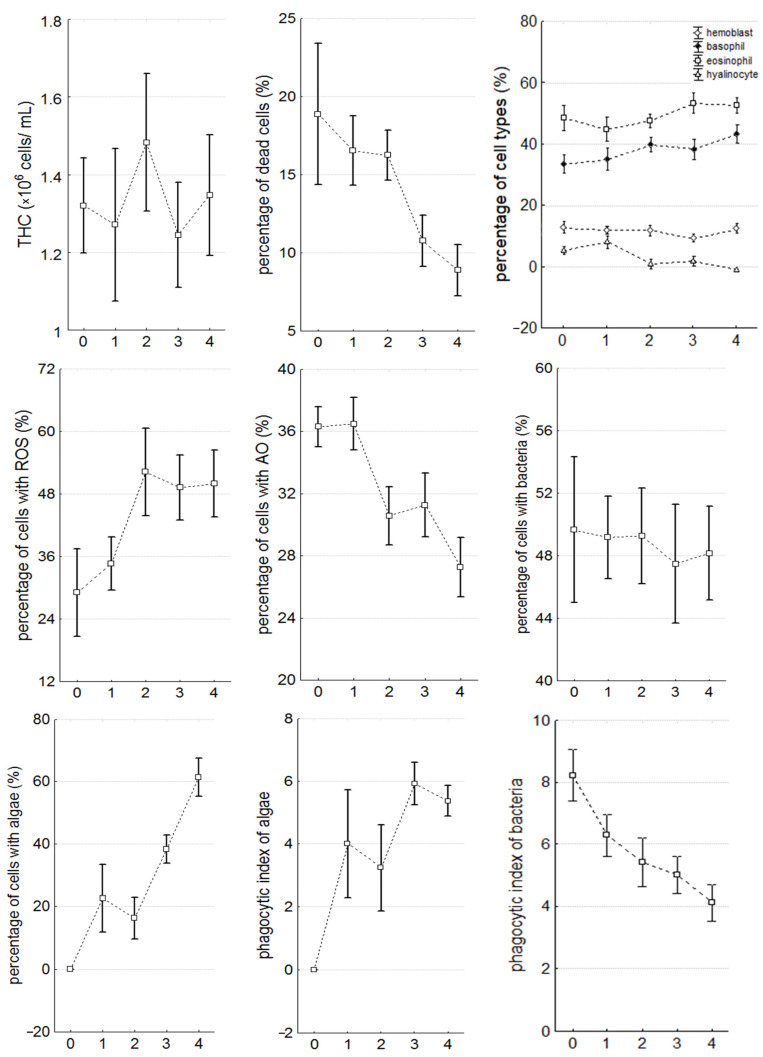
The parameters of the hemocytes during the progression of infection (THC—total hemocyte count, ROS—reactive oxygen species, AO—acridine orange). The values are presented as mean ± 95% confidence interval.

**Figure 4 microorganisms-12-00997-f004:**
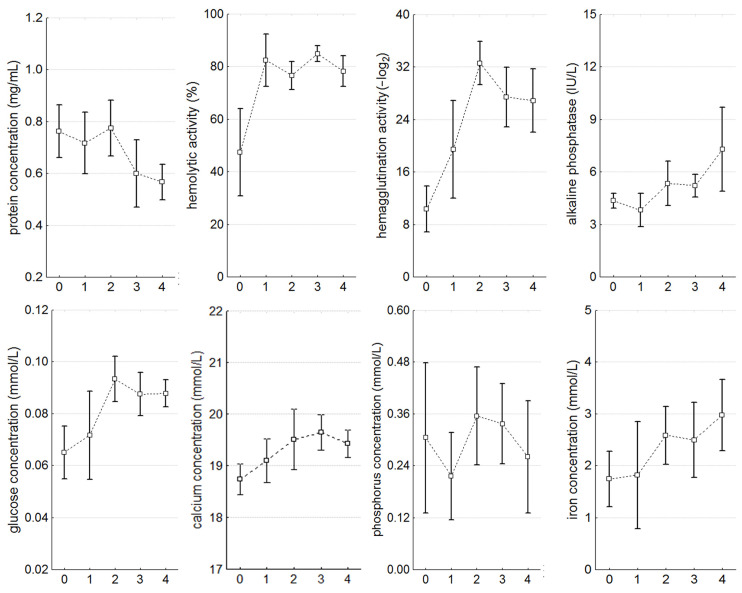
Parameters of the plasma during the progression of infection. The values are presented as mean ± 95% confidence interval.

**Figure 5 microorganisms-12-00997-f005:**
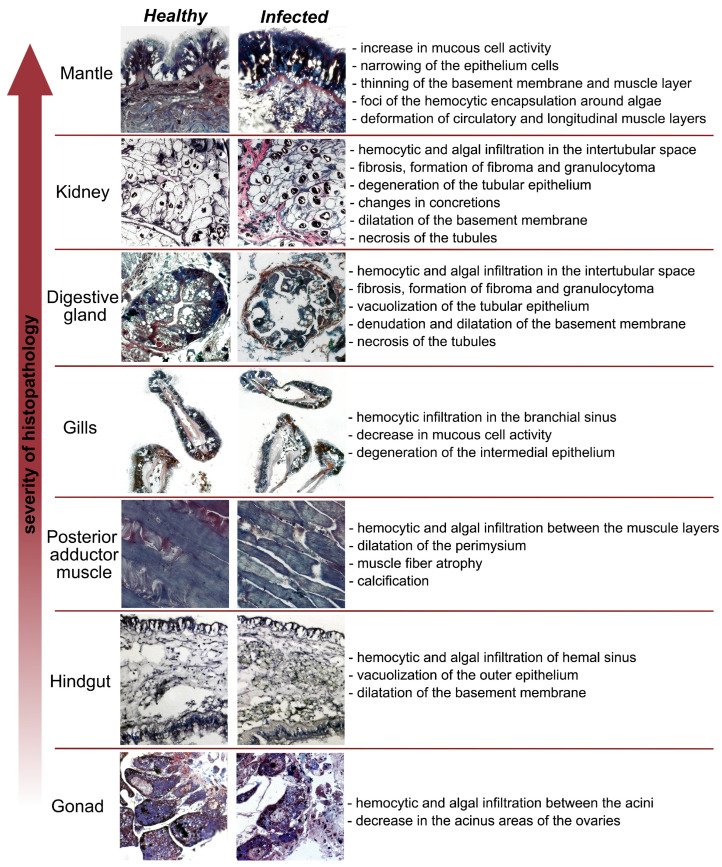
Histopathologic alterations in organs of the *M. kurilensis* at stage 4 of infection.

**Figure 6 microorganisms-12-00997-f006:**
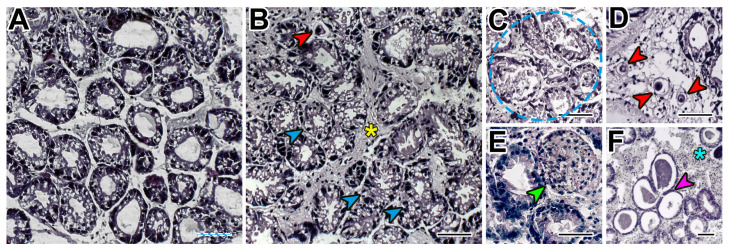
Morphology of a healthy (**A**) and pathologically changed (**B**–**F**) digestive gland (red arrow—*Nematopsis* sp., blue arrow—hypervacuolization, yellow asterisk—fibrosis, blue dotted line—necrotic area, green arrow—granulocytoma, blue asterisk—hemocyte infiltration, purple arrow—predominance of holding (I type) tubules). Scale bars: 50 μm.

**Figure 7 microorganisms-12-00997-f007:**
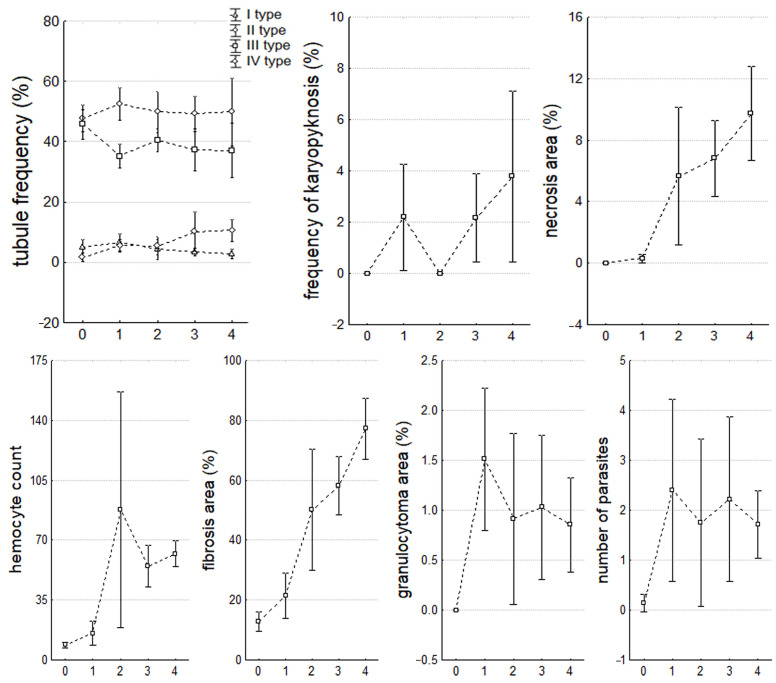
Parameters of digestive glands during the progression of infection. The values are presented as mean ± 95% confidence interval.

**Figure 8 microorganisms-12-00997-f008:**
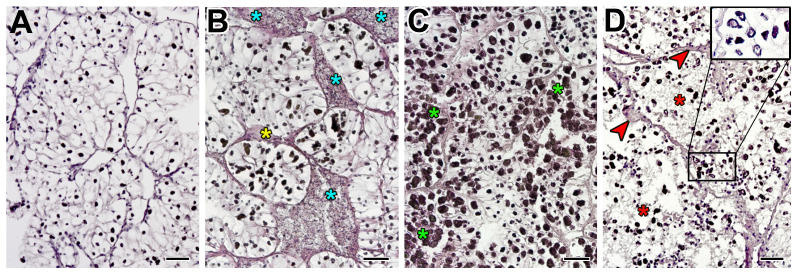
Representative micrographs of a healthy (**A**) and pathologically changed (**B**–**D**) nephridia (yellow asterisk—fibrosis, blue asterisk—hemocyte infiltration, green asterisk—increase in the number and size of concretions, red asterisk—necrotic area, red arrow—lamination and dilatation of the tubular basement membrane, black rectangle—alteration in the shape and density of concretions). Scale bars: 50 μm.

**Figure 9 microorganisms-12-00997-f009:**
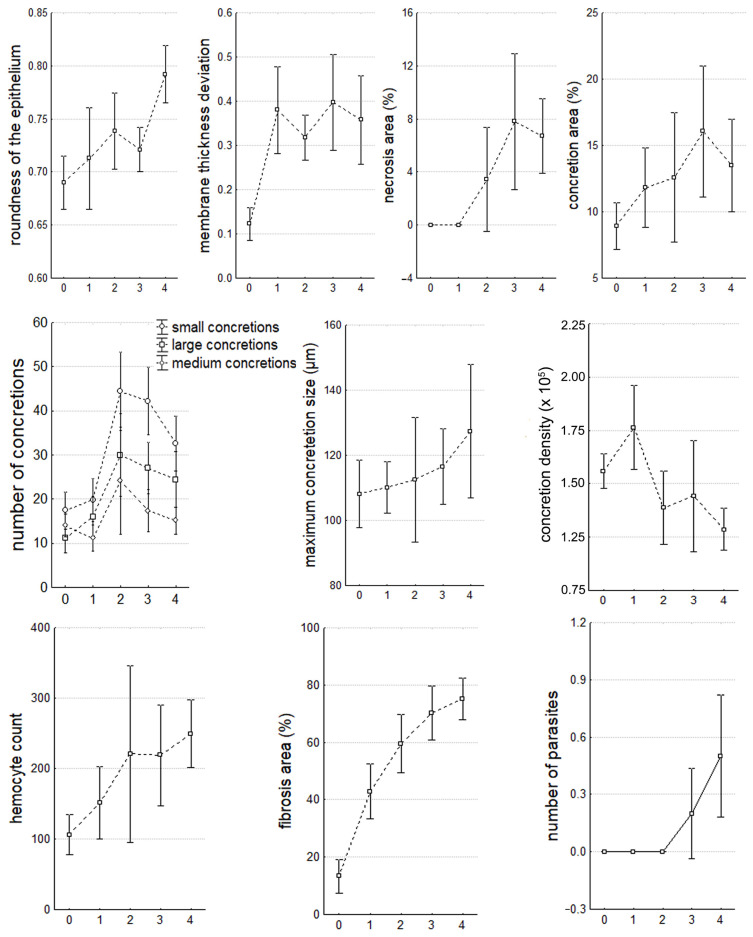
Parameters of nephridia during the progression of infection. The values are presented as mean ± 95% confidence interval.

**Table 1 microorganisms-12-00997-t001:** Comparative analysis of hemolymph parameters of *M. kurilensis* with varying stages of infection (THC—total hemocyte count, ROS—reactive oxygen species, AO—acridine orange, NS—no significant differences). Asterisks (*) indicate significant differences (Mann–Whitney U test, *p* < 0.05).

Parameter	Significance Level, *p*										
	Healthy–Infested	0–1	0–2	0–3	0–4	1–2	1–3	1–4	2–3	2–4	3–4
percentage of dead cells (%)	NS	NS	NS	0.016 *	0.009 *	NS	0.001 *	0.001 *	0.000 *	0.000 *	NS
percentage of basophils (%)	0.032 *	NS	0.050 *	NS	0.008 *	NS	NS	NS	NS	NS	NS
percentage of eosinophils (%)	0.050 *	NS	NS	0.042 *	NS	NS	0.008 *	0.012 *	0.028 *	0.035 *	NS
percentage of hyalinocytes (%)	0.000 *	NS	0.000 *	0.001 *	0.000 *	0.000 *	0.000 *	0.000 *	NS	NS	0.016 *
percentage of cells with ROS (%)	0.000 *	NS	0.002 *	0.000 *	0.000 *	NS	NS	0.011 *	NS	NS	NS
percentage of cells with AO (%)	0.001 *	NS	0.002 *	0.007 *	0.000 *	0.004 *	0.011 *	0.000 *	NS	NS	0.050 *
phagocytic index of bacteria	0.000 *	NS	0.000 *	0.002 *	0.000 *	0.023 *	0.020 *	0.018 *	NS	NS	NS
phagocytic index of algae	0.000 *	0.000 *	0.001 *	0.000 *	0.000 *	NS	0.029 *	0.046 *	0.009 *	0.044 *	NS
percentage of cells with algae (%)	0.000 *	0.000 *	0.001 *	0.000 *	0.000 *	NS	0.029 *	0.001 *	0.000 *	0.000 *	0.000 *
hemolytic activity (%)	0.000 *	0.006 *	0.008 *	0.000 *	0.021 *	NS	NS	NS	0.032 *	NS	0.029 *
hemagglutination activity (−log_2_)	0.000 *	NS	0.000 *	0.000 *	0.000 *	0.011 *	NS	NS	NS	NS	NS
glucose concentration (mmol/L)	0.001 *	NS	0.001	0.002 *	0.000 *	0.010 *	0.021 *	0.004 *	NS	NS	NS
calcium concentration (mmol/L)	0.004 *	NS	NS	0.002 *	0.003 *	NS	NS	NS	NS	NS	NS

**Table 2 microorganisms-12-00997-t002:** Comparative analysis of digestive glands and nephridia parameters of *M. kurilensis* with varying stages of infection (THC—total hemocyte count, ROS—reactive oxygen species, AO—acridine orange, IS—interstitial space). Asterisks (*) indicate significant differences (Mann–Whitney U test, *p* < 0.05).

Digestive Gland Parameters	Significance Level, *p*										
	Healthy–Infested	0–1	0–2	0–3	0–4	1–2	1–3	1–4	2–3	2–4	3–4
percentage of IV type tubules (%)	0.00 *	0.00 *	0.05 *	0.02 *	0.00 *	NS	NS	0.03 *	NS	0.05 *	NS
fibrosis area in the IS (%)	0.00 *	NS	0.02 *	0.00 *	0.00 *	NS	0.00 *	0.00 *	NS	NS	0.03 *
hemocyte count in the IS	0.00 *	NS	0.01 *	0.00 *	0.00 *	0.03 *	0.00 *	0.00 *	NS	NS	NS
parasite number	0.01 *	NS	NS	NS	0.00 *	NS	NS	NS	NS	NS	NS
necrosis area (%)	0.00 *	NS	0.00 *	0.00 *	0.00 *	0.01 *	0.00 *	0.00 *	NS	0.05 *	NS
frequency of kariopiknosis (%)	0.05 *	NS	-	0.04 *	0.01 *	NS	NS	NS	0.05 *	0.01 *	NS
granulocytoma area (%)	0.00 *	0.00 *	0.01 *	0.00 *	0.00 *	NS	NS	NS	NS	NS	NS
Nephridium Parameters											
max. membrane thickness (µm)	NS	NS	NS	0.02 *	0.01 *	NS	NS	NS	0.00 *	NS	0.00 *
membrane thickness deviation (µm)	0.00 *	0.00 *	0.00 *	0.00 *	0.00 *	NS	NS	NS	NS	NS	NS
epithelial roundness	0.01 *	NS	0.05 *	NS	0.00 *	NS	NS	0.03 *	NS	0.04 *	0.01 *
concretions number	0.00 *	NS	0.05 *	0.01 *	0.00 *	NS	NS	NS	NS	NS	0.04 *
concretions roundness	0.00 *	NS	0.00 *	0.01 *	0.04 *	NS	NS	NS	NS	NS	NS
concretions density	0.05 *	NS	0.09	0.05 *	0.00 *	0.02 *	0.01 *	0.00 *	NS	NS	NS
parasite number	NS	NS	NS	NS	0.02 *	-	NS	0.01 *	NS	0.02 *	NS
necrosis area (%)	0.01 *	0.02 *	NS	NS	NS	NS	NS	0.01 *	NS	NS	NS
fibrosis area in the IS (%)	0.00 *	0.00 *	0.00 *	0.00 *	0.00 *	NS	0.01 *	0.00 *	NS	0.03 *	NS
hemocyte count in the IS	0.00 *	NS	0.03 *	0.04 *	0.00 *	NS	NS	0.01 *	NS	NS	NS

## Data Availability

Data are contained within the article.

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
