# Peer review of "Study of the Impact of the Parasitic Microalgae Coccomyxa parasitica on the Health of Bivalve Modiolus kurilensis"

_microorganisms, 2024, doi:10.3390/microorganisms12050997_

Round 1
Reviewer 1 Report
Comments and Suggestions for Authors
Dear Editor and authors, this manuscript compares different types of parasitism to the parasitism described by Coccomyxa parasitica in the bivalve Modiolus kurilensis. I recommend the publication as the manuscript presents promising results that can be used in the future for monitoring. I suggest small revisions regarding the uses of rate nomenclature. Authors must use the pre-established nomenclature rules: Every time a taxon is presented for the first time it is necessary to include the author and year of the species, followed by the order and family: for example, for Modiolus kurilensis - Modiolus kurilensis Bernard, 1983 (Bivalvia: Mytilidae). Authors must also standardize whether taxa will be cited in the text with the first epithet abbreviated or in full.

Author Response
Response to Reviewer 1
We thank the Reviewer for taking the time to review our manuscript and providing valuable recommendations. Please find the detailed responses below and in the PDF review.
The authors revised the use of rate nomenclature. We also included the author and year of appearance of the species, followed by order and family in accordance with the Reviewer's recommendations. We standardized the citation of taxa and added full citation of the first epithet.
A revised version of the manuscript with corrections has been resubmitted.

Reviewer 2 Report
Comments and Suggestions for Authors
The study of the impact of the parasitic microalgae Coccomyxa parasitica on the health of the bivalve Modiolus kurilensis falls within the topic of this Journal. Furthermore, the shown results could indeed be used in the following monitoring programs as the stages of the invasion with the microalgae are shown in details and Figures are used to describe the visual invasion with the green microalgae. But, I strongly suggest the authors of the paper to consider rewriting the Discussion section: shorten it, and make it more readable. Explain your results and compare with other REVELANT studies on that specific topic. The Discussion section is too broad and part of it can very well be left out, or moved to Introduction section or just excluded from the paper and maybe used in the future for writing a Review on the topic. In the Discussion section there is no number/ data from this study compared to other previously mentioned study – please consider adding some concrete comparing. Also, English language throughout the paper should be rechecked. Therefore, I think this manuscript is publishable with major revision as suggested.
Specific comments for the authors are listed herein:
Abstract:
Line 18: Please consider using the appropriate term for “invertebrate`s kidney” to date = nephridia. And rename throughout the paper.
Keywords:
Line 24 comma is not needed between “immune disease”
Introduction:
Line 51-56 Please add citations regarding these sentences.
Line 55-57 Please define the term “massive invasion” and “weak infiltration”, add concentration if possible.
Line 60 It has been found…. sentence is missing citation.
Line 68 Please delete the term “at all” since it is not needed.
Materials and Methods:
Line 99-101 “Of” is not needed between the molarity and the chemical it is referring to, thus please rewrite all the molarity as the following example: 460 mM NaCl… You wrote it ok in the Line 148-149.
Line 102 Add “space” between 800 and “g” (in the Line 162 you wrote it right).
Line 102 Rephrase “Then” with “After centrifugation…”
Line 117 Please consider adding a Figure showing which cells you counted under the microscope (hemoblasts, agranulocytes, etc…) since it could be of use for further research.
Line 119 Note that the Latin words are always written in the italic: in vitro, in vivo, ex situ… Please correct throughout the paper. Also Lines 641 and 651
Line 122 FITC – it is the 1st time of mention, thus here should state the whole name of the method not just the abbreviation, that can be used further in the section.
Line 127-129 Please add two mathematical equations for the rations you counted.
Line 140 Is it that the coverslip had adherent living cells attached onto it? If so, please replace “with” with “coverslip containing…”.
Line 149 Correct the “15 mM CaCl2”.
Line 151 I would suggest to write the exponentiation as follows: 6x106 cells/ml.
Line 159 Again “of” is not neede.
Line 160 “were incubated”, not was since you are talking about tubes (plural).
Line 166-169 The section “biochemical analysis” contains too little data. Should be rewritten, also what are “standard settings”?
Line 170-175 Add figure number to access this – is it Figure 1? If so, mention it in this section. Please note that everything you counted for via the microscope should also be photographed and assigned so that other laboratories throughout the globe can repeat you study.
Line 186 Is this the 1st time the PFA is mentioned? If so, whole word should be stated with the abbreviation. Also, I do not understand what does the verb “processed” stands for in this sentence, is it really necessary or van it be excluded? The embeddement in paraffin could be further explained with a sentence.
Line 189 Please call to the Figure where it can be seen.
Line 190 Add “on THE work…”
Line 210-212 Is this small section a mistake – please delete if so.
Results:
Line 223 Correct “compared TO WITH…” use just one, either “to” or “with”
Figure 1. D – it is not numerated in the text, it has been written wrong like “B”, please correct.
Line 160 Add “in THE homogenate” …
Line 245 It would be nice to have a figure of percentage of the different infection stager in the entire sample, perhaps in one Barr chart.
Line264-268 Should be stated “Figure 2”, not “Figure 1”.
Figure 2. Line 272 There might be an extra space between “the stage 3”, if so delete.
Line 279. “In THE individuals…” Also, not “here” but “there”.
Line 282 A verb is missing at the end of the sentence to make it more understandable, consider adding” was recorded” at the end of the sentence.
Line 308 “Virtually” – consider using another word or just without.
Line 322 “even” is not needed here.
Line 332 Since you mentioned the “digestive gland (DG) in just the sentence before you may write just “DG” at this point.
Line 333 Change “than of” to “compared to”.
Table 2. Check the exact brackets because here you have two of them opened and thus should be different, and all of them should be closed – and this is not the situation inhere.
Line 346 rephrase the “area of necrosis” to “necrotic area”.
Line 355 Healthy mollusks is shown in the Figure 6A not E.
Figure 8. I do not see the “E” as it is stated in the “Figure explanation” bellow the Figure.
Figure 9. “Parameters of kidney” à “Nephridia parameters”.
Discussion:
Line 402-406 Please rephrase the sentence it is too long and hardly trackable with the main point. Delete not necessary part, or just write two sentences.
Line 405 Replace “and” with “thus poisoning mollusks”.
Line 406-407 Please consider the following throughout the paper: when and organism Latin name is 1st time mentioned the whole name should be stated, further when mentioning only the abbreviated term is to be used – this is not the case in your Discussion. Please check and write correctly. Example: here P. abbreviate is used for the 1st time (if we exclude the “Abstract”), same states for the A. broughtonii – full name should be listed.
Line 408-479 This is a too long part of the Discussion section that has no comparison with your results – thus it is not relevant to “discuss” – can be used in the Introduction or just leave behind for a Review paper.
Line 414 it is not clear “which study” since there is no reference- are you talking about your study or of someone else – if so, please add citation. Please throughout the Discussion reconsider using the terms “in our study” od “In the study by…” because in a lot of parts in the Discussion section this is not clear.
Line 424 Add” In all OF…”
Line 469 MSX abbreviation is used the 1st time here – please write the full words.
Line 473 replace the verb “is” with “was found to be…”.
Line 475 not “is” but “are” (plural – DG and hemolymph).
Line 474 The authors can start the Discussion section here…
Line 482-485 Two sentences that follow each other start with the same “As the disease progressed” – rephrase, and please add a stage that is relevant for these statements.
Line 490 before this sentence you describe your results, and here it should be stated clearly that with this sentence you now are comparing to other studies, so please use “In the study by...” or “In another research…”…
Line 499 after Perkinsus – citation is missing.
Line 505 The latin name is sometimes written in italic other times not: please see Line 505 “Perkinsus cells” and compare to Line 499 – this is to be checked throughout the Discussion section, and not only with “Perkinsus”.
Line 519 Has repetitions “a quantitative” – delete.
Line 521 Needs citation – more than one where the biomonitoring of what you are saying is used. Also, in the next sentence “to this” is not needed.
Line 522 A dot is put before the reference brackets.
Line 527 In the body of which organism?
Line 529 Suggestion to rephrase “For instance”…
Line 544 rephrase “catastrophic” or delete… or led to… something: a process in an organism.
Line 546. “But CAN also…”
Line 547-550 The “kidney” function should be explained through its role in invertebrate species as nephridia.
Line 548-549 reference needed.
Line 550 as bioindicators for what? Usually we call bioindicators the organisms, and biomarkers molecules that are elevated or downregulated in stressful environmental conditions.
Line 566 1st - st should be written as “superscript” also Line 619
Line 576 the results of your study that you mention here for sure can be compared to other studies, and it is not done at this point.
Line 580 reference missing
Line 592-593 Not sure why this sentence is relevant at this point, it is not necessary at all.
Line 597 NBT test?
Line 631 I would suggest to use another verb here not “may resort”.
Line 631-690 Again little to no comparison to the results obtained in this study, and the sentences that refer to the results are very broad – no specific data comparison between this study and another one is done.
Line 651-653 IAP – full naming when you mention it 1st time not the 2nd.
Line 666 T gigas – is this 1st time of mentioning? If so, add full Latin name.
Line 671 “among the other things” – delete
Line 681 – How is all this data you mentioned from other previously mentioned studies compared to your research?
Conclusions:
Line 704 “various invasions” – of what, which organisms – it is missing… of pathogens.
Line 706 “… control IS RELIES???” – choose one.
References:
The Latin names of the organisms should be written in italic throughout the References section.
Comments on the Quality of English LanguageEnglish language needs revision throughout the paper.
Author Response
Response to Reviewer 2
General comments
….I strongly suggest the authors of the paper to consider rewriting the Discussion section: shorten it, and make it more readable. Explain your results and compare with other REVELANT studies on that specific topic. The Discussion section is too broad and part of it can very well be left out, or moved to Introduction section or just excluded from the paper and maybe used in the future for writing a Review on the topic. In the Discussion section there is no number/ data from this study compared to other previously mentioned study – please consider adding some concrete comparing. Also, English language throughout the paper should be rechecked. Therefore, I think this manuscript is publishable with major revision as suggested.
Answer: The authors are grateful to the Reviewer for the kind review and constructive criticism. In the corrected version of this work, we took into account all the comments and suggestions. The English writing has been checked and corrected by the Language Editing Service. We believe that taking into account the Reviewer's recommendations has significantly improved the quality of the manuscript. The corrections described below have been made to the manuscript.
Unfortunately, there are no studies on the quantitative assessment of the development of infection caused by microalgae Coccomyxa and the response of bivalves. Currently, 16 articles have been published on the study of microalgae in bivalves. 11 articles are devoted to the description of microalgae (Zuykov et al., 2014; 2018; Sokolnikova et al., 2016; 2023; Syasina et al., 2012; Vaschenko et al., 2013; Vazquez et al., 2010; Crespo et al., 2009; Rodríguez et al., 2008; Mortensen et al., 2005; Gray et al., 1999) and their life cycle (Sokolnikova et al., 2016; 2023; Belzile, Gosselin, 2015), of which only 9 articles show a phylogenetic analysis of microalgae. 11 articles are devoted to the description of organ histopathologies (Zuykov et al., 2014; 2018; Sokolnikova et al., 2016; 2023; Syasina et al., 2012; Vazquez et al., 2010; Crespo et al., 2009; Rodríguez et al., 2008; Mortensen et al., 2005; Gray et al., 1999). 6 articles are devoted to the analysis of shell deformation in infected mollusks (Zuykov et al., 2014; 2018; 2020; Sokolnikova et al., 2016; 2023; Gray et al., 1999). And only one review summarizing the available knowledge about Coccomyxa and its owners. In the discussion, we compare the results of our work with these articles. We also attempted to comprehensively compare the response of mollusks to a rather poorly studied microalgae with available data in the literature on other pathogens in order to understand the specificity of the response of mollusks to unique phototrophic parasites. This attempt is probably the reason for the redundancy of the Discussion chapter. Because of the lack of similar works on microalgae, we also did not dare to compare our data with data from works on other mollusk parasites. The discussion has been condensed as recommended by the Reviewer (because the introduction should not be overloaded and one of authors of this manuscript has already been published (Sokolnikova, 2022)).
Specific comments for the authors are listed herein:
Abstract:
Line 18: Please consider using the appropriate term for “invertebrate`s kidney” to date = nephridia. And rename throughout the paper.
Answer: The authors replaced “kidney” with “nephridia” throughout the text.
Keywords:
Line 24 comma is not needed between “immune disease”
Answer: The authors have made corrections.
Introduction:
Line 51-56 Please add citations regarding these sentences.
Answer: The citations has been added (line 60).
Line 55-57 Please define the term “massive invasion” and “weak infiltration”, add concentration if possible.
Answer: The cited work does not contain a definition of the concepts of “massive” or “weak infiltration”, as well as numerical data.
Line 60 It has been found…. sentence is missing citation.
Answer: The citation has been added (line 66).
Line 68 Please delete the term “at all” since it is not needed.
Answer: The authors have made corrections.
Materials and Methods:
Answer: The authors agree with the Reviewer. All necessary corrections mentioned below by the reviewer have been made.
Line 99-101 “Of” is not needed between the molarity and the chemical it is referring to, thus please rewrite all the molarity as the following example: 460 mM NaCl… You wrote it ok in the Line 148-149.
Line 102 Add “space” between 800 and “g” (in the Line 162 you wrote it right).
Line 102 Rephrase “Then” with “After centrifugation…”
Line 119 Note that the Latin words are always written in the italic: in vitro, in vivo, ex situ… Please correct throughout the paper. Also Lines 641 and 651
Line 122 FITC – it is the 1st time of mention, thus here should state the whole name of the method not just the abbreviation, that can be used further in the section.
Line 140 Is it that the coverslip had adherent living cells attached onto it? If so, please replace “with” with “coverslip containing…”.
Line 149 Correct the “15 mM CaCl2”.
Line 151 I would suggest to write the exponentiation as follows: 6x106 cells/ml.
Line 159 Again “of” is not neede.
Line 160 “were incubated”, not was since you are talking about tubes (plural).
Line 189 Please call to the Figure where it can be seen.
Line 190 Add “on THE work…”
Line 210-212 Is this small section a mistake – please delete if so.
Line 117 Please consider adding a Figure showing which cells you counted under the microscope (hemoblasts, agranulocytes, etc…) since it could be of use for further research.
Answer: A detailed description and illustration of the types of hemocytes in Modiolus kurilensis has already been published in the work of Sokolnikova and colleagues (2022). We have added the reference to the text of this manuscript (line 123).
Line 127-129 Please add two mathematical equations for the rations you counted.
Answer: We considered that an introduction of fairly simple formulas describing the proportion of phagocytes from all hemocytes would complicate the perception of the text, so we made adjustments to the concepts.
Line 166-169 The section “biochemical analysis” contains too little data. Should be rewritten, also what are “standard settings”?
Answer: Biochemical analysis in this work concluded of placing a test tube with plasma in a biochemical analyzer, selecting the parameter in the analyzer program settings that needs to be measured and starting this process (total protein, glucose, phosphorus, calcium, magnesium, iron and alkaline phosphatase). No additional manipulations were required.
Line 170-175 Add figure number to access this – is it Figure 1? If so, mention it in this section. Please note that everything you counted for via the microscope should also be photographed and assigned so that other laboratories throughout the globe can repeat you study.
Answer: In this case, we recorded the detected signs in the laboratory journal. After this, we counted the number of microalgae. Next, the results of both procedures were compared. We believe that adding a link in the “Materials and Methods” chapter to a figure from the “Results” chapter is incorrect, since according to the journal rules, the “Materials and Methods” chapter must contain a description of the methods.
Line 186 Is this the 1st time the PFA is mentioned? If so, whole word should be stated with the abbreviation. Also, I do not understand what does the verb “processed” stands for in this sentence, is it really necessary or van it be excluded? The embeddement in paraffin could be further explained with a sentence.
Answer: The first mention of PFA is on line 103. Samples were dehydrated through a progressive series of ethanol, infiltrated by xylene, and embedded in paraffin (this clarification has been added).
Results:
Answer: The authors agree with the Reviewer. All necessary corrections mentioned below by the reviewer have been made.
Line 223 Correct “compared TO WITH…” use just one, either “to” or “with”
Figure 1. D – it is not numerated in the text, it has been written wrong like “B”, please correct.
Line 160 Add “in THE homogenate” …
Line264-268 Should be stated “Figure 2”, not “Figure 1”.
Figure 2. Line 272 There might be an extra space between “the stage 3”, if so delete.
Line 279. “In THE individuals…” Also, not “here” but “there”.
Line 282 A verb is missing at the end of the sentence to make it more understandable, consider adding” was recorded” at the end of the sentence.
Line 308 “Virtually” – consider using another word or just without.
Line 322 “even” is not needed here.
Line 332 Since you mentioned the “digestive gland (DG) in just the sentence before you may write just “DG” at this point.
Line 333 Change “than of” to “compared to”.
Table 2. Check the exact brackets because here you have two of them opened and thus should be different, and all of them should be closed – and this is not the situation inhere.
Line 346 rephrase the “area of necrosis” to “necrotic area”.
Line 355 Healthy mollusks is shown in the Figure 6A not E.
Figure 8. I do not see the “E” as it is stated in the “Figure explanation” bellow the Figure.
Figure 9. “Parameters of kidney” à “Nephridia parameters”.Line 245 It would be nice to have a figure of percentage of the different infection stager in the entire sample, perhaps in one Barr chart.
Answer: Since the ratio of the number of individuals at different stages is approximately the same (14.6, 16.7 and 18.7%), except of stages 2 and 3 (25% each), the authors considered it unnecessary to add this illustration.
Discussion:
Answer: The authors agree with the Reviewer. All necessary corrections mentioned below by the reviewer have been made.
Line 402-406 Please rephrase the sentence it is too long and hardly trackable with the main point. Delete not necessary part, or just write two sentences.
Line 405 Replace “and” with “thus poisoning mollusks”.
Line 406-407 Please consider the following throughout the paper: when and organism Latin name is 1st time mentioned the whole name should be stated, further when mentioning only the abbreviated term is to be used – this is not the case in your Discussion. Please check and write correctly. Example: here P. abbreviate is used for the 1st time (if we exclude the “Abstract”), same states for the A. broughtonii – full name should be listed.
Line 414 it is not clear “which study” since there is no reference- are you talking about your study or of someone else – if so, please add citation. Please throughout the Discussion reconsider using the terms “in our study” od “In the study by…” because in a lot of parts in the Discussion section this is not clear.
Line 424 Add” In all OF…”
Line 469 MSX abbreviation is used the 1st time here – please write the full words.
Line 473 replace the verb “is” with “was found to be…”.
Line 475 not “is” but “are” (plural – DG and hemolymph).
Line 474 The authors can start the Discussion section here…
Line 482-485 Two sentences that follow each other start with the same “As the disease progressed” – rephrase, and please add a stage that is relevant for these statements.
Line 490 before this sentence you describe your results, and here it should be stated clearly that with this sentence you now are comparing to other studies, so please use “In the study by...” or “In another research…”…
Line 499 after Perkinsus – citation is missing.
Line 505 The latin name is sometimes written in italic other times not: please see Line 505 “Perkinsus cells” and compare to Line 499 – this is to be checked throughout the Discussion section, and not only with “Perkinsus”.
Line 519 Has repetitions “a quantitative” – delete.
Line 521 Needs citation – more than one where the biomonitoring of what you are saying is used. Also, in the next sentence “to this” is not needed.
Line 522 A dot is put before the reference brackets.
Line 527 In the body of which organism?
Line 529 Suggestion to rephrase “For instance”…
Line 544 rephrase “catastrophic” or delete… or led to… something: a process in an organism.
Line 546. “But CAN also…”
Line 547-550 The “kidney” function should be explained through its role in invertebrate species as nephridia.
Line 548-549 reference needed.
Line 550 as bioindicators for what? Usually we call bioindicators the organisms, and biomarkers molecules that are elevated or downregulated in stressful environmental conditions.
Line 566 1st - st should be written as “superscript” also Line 619
Line 576 the results of your study that you mention here for sure can be compared to other studies, and it is not done at this point.
Line 580 reference missing
Line 592-593 Not sure why this sentence is relevant at this point, it is not necessary at all.
Line 651-653 IAP – full naming when you mention it 1st time not the 2nd.
Line 666 T gigas – is this 1st time of mentioning? If so, add full Latin name.
Line 671 “among the other things” – delete
Line 408-479 This is a too long part of the Discussion section that has no comparison with your results – thus it is not relevant to “discuss” – can be used in the Introduction or just leave behind for a Review paper.
Answer: This manuscript is devoted to describing the effects of microalgae on the health of bivalves. To understand and prove the parasitic effect of microalgae, comparison with works on other bivalves pathogens is necessary. In this part, the authors discuss the histopathology that can be caused by various pathogens, including Coccomyxa. And this part begins with the sentence “It has been found that Coccomyxa infection in Mytilidae family members causes microalgae infiltration in all organs, but with a predominance in the mantle, gonad, kidneys and DG [6–9,14,16,17,36,37]." The authors have shortened this part as recommended by the Reviewer.
Line 597 NBT test?
Answer: The NBT test is described in Materials and Methods on line 139.
Line 631-690 Again little to no comparison to the results obtained in this study, and the sentences that refer to the results are very broad – no specific data comparison between this study and another one is done.
Answer: Similar studies performed on shellfish infected with microalgae have not been published. Our work is the first. The discovered reliable dynamics of changes in immune parameters during the development of infection revealed similarities with similar works on mollusks infected with the most dangerous pathogens (Perkinsus, Haplosporidium, Bonamia, Marteilia). Which raises concerns about the impact of Coccomyxa on bivalves in the future.
Line 681 – How is all this data you mentioned from other previously mentioned studies compared to your research?
Answer: In this part of the “Discussion” chapter, we made an attempt to explain the reason for the changes in the content of calcium, iron, glucose and other components in the plasma of infected bivalves, comparing them with the facts of other works. For example, “The increase in glucose may be a consequence of the transport of this metabolite from microalgae. Similar changes were observed in…”
Conclusions:
Line 706 “… control IS RELIES???” – choose one.
Answer: The authors have made corrections.
Line 704 “various invasions” – of what, which organisms – it is missing… of pathogens.
Answer: Our conclusion concerns result of this work, its usefulness and prospects. The sentence on line 704 only “High mortality rate of bivalves because of various invasions is a major issue in aquaculture that leads to significant economic loss in the industry. ” points to an existing problem, which is the first in aquaculture because of a lack of knowledge (which the authors declare throughout the entire chapter “Discussion”, including pointing out the most dangerous pathogens by species, and comparing their negative impact on host with the impact of Coccomyxa).
References:
The Latin names of the organisms should be written in italic throughout the References section.
Answer: The authors have made corrections.
Round 2
Reviewer 2 Report
Comments and Suggestions for Authors
after the author's correction I am satisfied with all the corrections taken into account and comments, thus I would suggest publishing the Article.